# Disturbances of the Lung Glutathione System in Adult Guinea Pigs Following Neonatal Vitamin C or Cysteine Deficiency

**DOI:** 10.3390/antiox12071361

**Published:** 2023-06-29

**Authors:** Vitor Teixeira, Ibrahim Mohamed, Jean-Claude Lavoie

**Affiliations:** 1Department of Nutrition, Université de Montréal, Montréal, QC H3T 1C5, Canada; vitor.teixeira.nascimento@umontreal.ca (V.T.); ibrahim.mohamed@umontreal.ca (I.M.); 2Department of Pediatrics-Neonatology, CHU Sainte-Justine, Université de Montréal, Montréal, QC H3T 1C5, Canada

**Keywords:** thrifty phenotype hypothesis, developmental origins of health and disease (DOHaD), lungs, bronchopulmonary dysplasia, ascorbic acid, glutathionylation, glutaredoxin, glutathione reductase, lung disease, antioxidant deficiencies

## Abstract

In premature infants receiving parenteral nutrition, oxidative stress is a trigger for the development of bronchopulmonary dysplasia, which is an important factor in the development of adult lung diseases. Neonatal vitamin C and glutathione deficiency is suspected to induce permanent modification of redox metabolism favoring the development of neonatal and adult lung diseases. A total of 64 3-day-old guinea pigs were fed an oral diet that was either complete or deficient in vitamin C (VCD), cysteine (CD) (glutathione-limiting substrate) or both (DD) for 4 days. At 1 week of age, half of the animals were sacrificed while the other started a complete diet until 12 weeks of age. At 1 week, the decrease in lung GSH in all deficient groups was partially explained by the oxidation of liver methionine-adenosyltransferase. mRNA levels of kelch-like ECH-associated protein 1 (*Keap1*), glutathione-reductase (*Gsr*) and glutaredoxin-1 (*Glrx*) were significantly lower only in CD but not in DD. At 12 weeks, glutathione levels were increased in VCD and CD. *Keap1*, *Gsr* and *Glrx* mRNA were increased, while glutathione-reductase and glutaredoxin proteins were lower in CD, favoring a higher glutathionylation status. Both neonatal deficiencies result in a long-term change in glutathione metabolism that could contribute to lung diseases’ development.

## 1. Introduction

Very premature birth (≤32 weeks of gestation) is associated with several short- and long-term consequences over the development of several diseases. In neonatal life, infants born prematurely develop complications such as bronchopulmonary dysplasia [1], and they are at higher risk of developing asthma, chronic obstructive pulmonary disease (COPD) and other respiratory complications in adulthood [2,3], as well as other complications [1,2,4,5,6,7,8,9,10,11]. Despite the strong clinical evidence, it is challenging to seize what specific aspects of prematurity lead to the increased risk of neonatal and adult diseases. Because oxidative stress is a common feature of these neonatal and adult pulmonary diseases [4,12], the development of a permanent disruption of redox metabolism is suspected.

Besides early birth, at the time when organ development is underway, premature neonates are submitted to several medical interventions. One of them is parenteral nutrition. This mode of nutrition is administered to preterm neonates due to their intestinal immaturity. It is composed of mono- and oligomeric nutrients diluted in a solution that is infused intravenously. Because these nutrients are dissolved in the same solution, they can interact with each other. For instance, photo-excited riboflavin will interact with vitamin C and degrade the latter while also generating peroxides [13,14,15]. In fact, after 5 h of light exposure, half of vitamin C added to parenteral nutrition solution is degraded, and after 24 h, no vitamin C is found in the solution [13]. The pulmonary concentration of vitamin C is reduced in animals infused with parenteral nutrition devoid of photo-protection [16]. On the other hand, peroxides contaminating parenteral nutrition solutions inhibit glutathione synthesis by decreasing the conversion of methionine into cysteine [17,18], the limiting substrate for glutathione synthesis [19,20]. Thus, neonates receiving parenteral nutrition are at risk of low vitamin C intakes and low glutathione synthesis. These two molecules are important cellular antioxidants.

We have demonstrated that vitamin C and cysteine deficiencies lead to higher cytoplasmic Nrf2 in the neonatal liver of guinea pigs with decreased translocation of Nrf2 into the nucleus at adult life [7]. Nrf2 is the master regulator of the antioxidant response. Therefore, it is hypothesized that vitamin C and cysteine deficiencies lead to long-term effects on redox metabolism. These long-term effects could be due to the epigenetic programming induced by oxidative stress, as it has been demonstrated that it can induce DNA and histone hypermethylation and histone hypoacetylation [21,22]. We aimed to validate the effect of these neonatal deficiencies, independently of other parenteral nutrition components, over redox metabolism in adult guinea pigs’ lungs. We evaluated glutathione levels as well as the mRNA and protein levels of some key redox metabolism proteins. We hypothesized that an oxidative stress in neonatal life leads to an increase in antioxidant defenses at adulthood in guinea pigs.

Results confirm the strong impact of neonatal diets deficient in vitamin C and cysteine on redox metabolism. Compared to control animals, pulmonary glutathione was lower at the time of deficiencies (at 1 week of age) and higher 11 weeks after stopping the deficient diets. The cysteine-deficient diet induced a decrease in *Keap1*, *Gsr* and *Glrx* mRNA levels at 1 week of life, whereas they were higher at 12 weeks, which was accompanied by a long-term decrease in deglutathionylation enzymes, such as glutaredoxin and glutathione-reductase. Although males and females were used in this study, sex was not found to be a contributing factor to the assessed outcomes.

## 2. Materials and Methods

### 2.1. Experimental Procedures

Briefly, 3-day-old Hartley guinea pigs (Charles River Laboratories, St-Constant, QC, Canada) from both sexes were randomized into 4 groups (n = 8 males and 8 females per group) according to the diet given between day 3 and day 7 of life: (1) complete diet (**Control**); (2) vitamin C free diet (Vitamin C-deficient—**VCD**); (3) cysteine-free diet (cysteine-deficient diet—C**D**); (4) vitamin C and cysteine-deficient diet (double deficient—**DD**). After 4 days of receiving the aforementioned diets, half of the animals in each group were sacrificed (1-week-old animals), and the other half were transitioned into a complete standard guinea pig diet (2041-Teklad Global High Fiber Guinea Pig Diet; Harlan, Montréal, QC, Canada) until the age of 12 weeks. All sacrifices were performed under isoflurane anesthesia, and 12-week-old animal sacrifices were performed after a 16 h fasting period. During sacrifice, lungs and liver were collected and quickly frozen at −80 °C and kept at this temperature until biochemical determinations. The study experimental design, including diet composition, has been published before [7].

All procedures were in accordance with the Canadian Council on Animal Care guidelines, and the study protocol was approved by the Institutional Animal Care Committee of the CHU-Sainte-Justine Research Centre (protocol #739).

### 2.2. Biochemical Assessments

#### 2.2.1. GSH, GSSG and Ascorbate

First, 250 milligrams of lung tissue were homogenized in 5 volumes of 5% (*w*/*v*) metaphosphoric acid and centrifuged at 7200× *g*/3 min. The supernatant was conserved at −80 °C, and the pellet was used for protein quantification by the Bradford method [23]. GSH and GSSG were resolved in a P/ACE MDQ Capillary Electrophoresis System (Beckman Coulter, Mississauga, ON, Canada) in a boric acid 75 mM pH 8.4, bis-tris 25 mM buffer. Separation was performed in a 75 µm/50 cm silica capillary, 28 °C, 18 kV, and absorbance was measured at 200 nm [24,25,26]. Ascorbate was resolved in a 50 µM/45 cm silica capillary, 25 °C, 30 kV in an Agilent 7100 Capillary Electrophoresis System (Agilent Technologies, Mississauga, ON, Canada). Separation was performed in boric acid 200 mM and acetonitrile 20% *v/v* buffer, pH 9.6, and detection of ascorbate was performed by absorbance at 268 nm [7,21].

#### 2.2.2. RT-qPCR

In lung tissue, the expression of genes involved in redox metabolism was measured by reverse transcriptase quantitative polymerase chain reaction. Thirty milligrams of lungs were homogenized, and RNA was extracted using the AllPrep DNA/RNA/miRNA Universal Kit (#80224, Qiagen Inc., Toronto, ON, Canada). To avoid interference from genomic DNA, RNA was treated with DNase I (#EN0525, ThermoFisher Scientific, Mississauga, ON, Canada), followed by reverse transcriptase treatment (#1725038, Bio-Rad, Mississauga, ON, Canada). Finally, qPCR was performed with iTaq™ Universal SYBR^®^ Green Supermix (#1725121, Bio-Rad Laboratories Ltd., Mississauga, ON, Canada). Primer pairs were previously tested for guinea pig genes, and efficiency was accepted at 100 ± 10%. The primers used and their concentrations are described in Table 1. Vinculin was used as a housekeeping gene and relative expression was calculated with the ∆∆Ct method, where ∆Ct = (Ct_Gene_ − Ct _Vcl_); ∆∆Ct = ∆Ct_Expreimental_ − ∆_CtControl group_; and relative expression = 2^−∆∆Ct^.

#### 2.2.3. Keap1, GR and GRX Protein Levels

Fifty to one hundred milligrams of lung tissue were homogenized in 9 volumes of buffer (Tris-HCl 50 mM pH 7.5, NaCl 150 mM and EDTA 5 mM) and centrifuged at 1300× *g* for 30 s at room temperature. The supernatant was collected, the protein concentration was measured, and the aliquots were stored at −80 °C. For the measurement of Keap1 and GR, 30 µg of protein was used, and for GRX, 50 µg was used. The samples were properly diluted in Laemmli’s buffer, and the proteins were denaturated at 95 °C for 5 min. The proteins were resolved in a 4/12% acrylamide gel at 110 V for 150 min (Keap1 and GR) or 120 min (GRX) and then transferred to a PVDF membrane. The membranes were blocked with skim milk 5% diluted in PBS-Tween 20 0.2%, and they were incubated with primary antibodies overnight. The antibodies used were the following: Keap1—mouse anti-human KEAP1 monoclonal antibody (1F10B6, ThermoFisher Scientific, St-Laurent, QC, Canada) (1:1000); GR—mouse anti-human glutathione-reductase monoclonal antibody (sc-133245, Santa Cruz Biotechnology, Santa Cruz, CA, USA) (1:1000); and GRX—rabbit anti-human glutaredoxin 1 polyclonal antibody (ab45953, Abcam Plc, Toronto, ON, Canada) (1:250); the membranes were washed in PBS-Tween 20 0.05% and incubated for 1h at room temperature with goat anti-mouse IgG-HRP antibody (HAF007, R&D Systems, Minneapolis, MN, USA) (1:2500) for GR and Keap1 membranes or goat anti-rabbit IgG-HRP antibody (W4011, Promega, Madison, WI, USA) (1:2500) for GRX membranes. The proteins were detected by chemiluminescence.

#### 2.2.4. Biotin Switch and Methionine Adenosyltransferase (MAT) Oxidation

The method of Jaffrey was used, with some modifications [27]. One hundred milligrams of frozen liver was homogenized in 10 volumes of HEN buffer (Hepes-Na 250 mM pH 7.7, EDTA 1 mM, neocuproine 0.1 mM) and centrifuged at 2000× *g*/10 min/4 °C. The supernatant was separated, and 3-[(3-Cholamidopropyl)dimethylammonio]-1-propanesulfonate (CHAPS) was added to a final concentration of 0.4%. Three hundred microliters of this sample were diluted in 4 volumes of HENSM buffer (HEN buffer + SDS 2.5% and 20 mM methyl methanethiosulfonate (MMTS)) and incubated at 50 °C/20 min under continuous agitation. MMTS was removed from samples by protein filtering in centrifugal 3 KDa protein filtering units, and samples were washed twice with HENS buffer (HEN buffer + SDS 1%). Samples were incubated with 50 µL of tris (2-carboxyethyl)phosphine (TCEP) 50 mM for 1 h/RT and had the TCEP removed by centrifugal filtering like the previous step. Reduced cysteines were probed with biotin through incubation of samples with 120 µL of 5-[(3*aS*,4*S*,6*aR*)-2-oxo-1,3,3*a*,4,6,6*a*-hexahydrothieno [3,4-d]imidazol-4-yl]-*N*-[6-[3-(pyridin-2-yldisulfanyl)propanoylamino]hexyl]pentanamide (biotin-HPDP) 4 mM in dimethylformamide (DMF) and incubated for 1 h/RT. Biotin-HPDP was removed by centrifugal filtering with two washes in HENS buffer. One hundred microliters of the biotinylated sample was diluted in 200 µL of neutralisation buffer (Hepes-Na 20 mM pH 7.7, NaCl 100 mM, EDTA 1 mM, Triton X-100 0.5%) and 10 µL of streptavidin-agarose resin beads was added. Samples were incubated for 1 h/RT. The beads were washed with 500 µL of neutralisation buffer plus NaCl 600 mM and centrifuged at 1400× *g*/5 s /RT. The supernatant was discarded, and the wash was repeated 4 other times. Protein elution was performed by the addition of 100 µL of elution buffer (Hepes-Na 20 mM pH 7.7, NaCl 100 mM, EDTA 1 mM, 2-mercaptoethanol 100 mM) and 10 min incubation at RT. Samples were centrifuged at 1400× *g*/1 min/RT, and the supernatant was conserved at −80 °C. Unpurified biotinylated (total proteins) and purified biotinylated samples (oxidized proteins) were submitted to Western blotting. Ten micrograms of total proteins or 45 µL of the purified oxidized proteins in 5 µL of loader buffer were denatured at 95 °C/5 min and resolved in a 4/12% gel as previously published for acetyl-CoA carboxylase protein [7,25]. MAT was detected with mouse anti-human MAT Ia/IIa monoclonal antibody (sc-166452, Santa Cruz Biotechnology) (1:1000) for 1 h/room temperature, followed by goat anti-mouse IgG-HRP antibody (HAF007, R&D Systems) (1:2500) for 1h/room temperature. Antibodies were diluted in 2.5% *w/v* skim milk in phosphate-buffer-saline-Tween-20.

#### 2.2.5. Statistics

All data are presented as mean ± S.E.M. Data from the four groups were orthogonally compared by 3-way-ANOVA, considering vitamin C, cysteine and sex as contributing factors. An absence of significant interactions between vitamin C and cysteine was interpreted as independent effects, while a significant interaction as observed for GSH, ascorbate and MAT was interpreted as reaching a plateau. When a significant interaction between vitamin C and cysteine was observed, we performed a Dunnett’s post hoc test. This test compares the control group against all other groups. The significance threshold was set at *p* = 0.05.

## 3. Results

Bodyweight, growth and other baseline characteristics of these animals have been described before [7]. Briefly, all animals had similar bodyweights at baseline, and while controls and VCD had 10 ± 3% and 7 ± 1% weight gain during the 4 days of experimental diets, CD and DD groups did not gain weight (−3 ± 1% and −6 ± 1%, respectively). Lung weight per body weight (Figure 1) was higher in all deficient animals at 1 week of age (*p* < 0.05) from both sexes (Figure 1). However, this effect seems to be caused by the variations in bodyweight in these animals [7], as no differences in lung weight itself are observed among groups (F_(1,23)_ = 0.32, *p* = 0.940) at one week of age. At 12 weeks, lung weight per bodyweight was not different among groups, but it was higher in females (F_Sex(1,24)_ = 25.8; *p* < 0.001). However, an interaction between cysteine deficiency and sex was observed (F_Cys × sex(1,24)_ = 5.0, *p* < 0.05). While cysteine deficiency decreased lung weight per body weight in 12-week-old males, it increased it in females. No other effects of sex were statistically significant for the other variables neither at 1 week (F_Sex(1,23)_ ≤ 3.1) nor at 12 weeks (F_Sex(1,24)_ ≤ 2.8). Therefore all the following results represent both male and female animals together.

### 3.1. Lung GSH and GSSG Levels and Redox Potential

At neonatal age, GSH was decreased by −34% in all deficient groups, compared to the control (Figure 2A). Double deficient animals did not have a further decrease (F_VC × Cys (1,23) =_ 12.4, *p* < 0.01). Dunnet’s post hoc analysis showed that all groups have a similar decrease (VCD: −37%, CD: −34%; DD: −31%; *p* < 0.01). GSSG, on the other hand was only decreased in cysteine-deficient animals (F_Cys (1,23) =_ 12.6, *p* < 0.01), suggesting a decrease in total glutathione with no redox imbalance (Figure 2B). This is confirmed by the redox potential of glutathione that is only oxidized in vitamin C-deficient animals (VCD: +11.3 mV; *p* < 0.05) and not in cysteine-deficient animals (CD: +3.6 mV; *p* = 0.996) (Figure 2C).

At 12 weeks of age, GSH and GSSG were increased in all animals submitted to a deficiency early in life (Figure 2A,B). However, the effects were additive for GSSG (F_VC (1,24)_ = 24.4, *p* < 0.001; F_Cys (1,24)_ = 9.0, *p* < 0.01) but not for GSH, as evidenced by the significant interaction (F_VC × Cys (1,24)_ = 4.6, *p* < 0.05) and the significant effects when the DD group is not included in the analysis (F_VC (1,18)_ = 10, *p* < 0.01; F_Cys (1,18)_ = 4.9, *p* < 0.05). No differences in redox potential were observed (F_(1,24)_ = 0.8) (Figure 2C).

### 3.2. Pulmonary Ascorbate

At 1 week of age, both vitamin C and cysteine deficiencies decreased lung ascorbate (F_VC (1,23)_ = 4.8, *p* < 0.05; F_Cys (1,23)_ = 10.8, *p* < 0.01). A further decrease was not observed in the DD group. No differences were observed at 12 weeks of age (Figure 2D).

### 3.3. Lung Gene Expression

No statistical differences were observed in the expression of *Nfe2l2*, *Gclc* or *Gclm* (Figure 3). However, a significant positive correlation between *Nfe2l2* and *Gclc* expression was observed (Gclc_mRNA_ = 1.0362 × (Nfe2l2_mRNA_) + 0.123; *r*^2^ = 0.59; *p* < 0.001), confirming the role of Nrf2 in inducing *GCLC* expression. A similar correlation was also observed for *Nfe2l2* and *Gclm*, suggesting that Nrf2 plays a role in *GCLM* expression. However, according to the *r*^2^ value, Nrf2 did not seem to be the preponderant factor in *Gclm* expression, as it only contributed to 16% of the variation observed in the *Gclm* data (Gclm_mRNA_ = 0.91 × (Nfe2l2_mRNA_) + 0.51; *r*^2^ = 0.16; *p* < 0.01).

Neonatal cysteine deficiency led to a decrease in mRNA levels of *Keap1*, *Gsr* and *Glrx* (*p* < 0.05) at one week of life, while an increase in their gene expression was observed at 12 weeks of age (*p* < 0.05) (Figure 3). The expression of these genes was highly and positively correlated (Figure 4). Vitamin C deficiency did not significantly affect the expression of any of the assessed genes.

### 3.4. Protein Levels of Keap1, GR and GRX

At one week of life, no significant changes were observed for Keap1, GR or GRX protein levels (F_(1,23)_ < 3.4) (Figure 5). At 12 weeks of life, no significant differences were observed for Keap1 protein levels (F_(1,24)_ < 3.0). A significant decrease in GR protein levels was observed in all experimental groups. An interaction between vitamin C and cysteine deficiencies is also observed (F_VC × Cys (1,24)_ = 6.8, *p* < 0.05), and Dunnet’s post hoc test revealed that each of the groups that received a deficient diet during neonatal life had significantly lower GR levels (*p* < 0.05). A similar decrease was also observed in GRX protein levels in cysteine-deficient groups at 12 weeks of life (F_Cys (1,17)_ = 5.1, *p* < 0.05).

### 3.5. Hepatic Methionine-Adenosyltransferase (MAT) Oxidation

At one week of life, liver MAT was significantly more oxidized in all deficient groups compared to the control (Figure 6A). Following a significant interaction between vitamin C and cysteine deficiencies (F_VC × Cys (1,23)_ = 22.4, *p* < 0.001), Dunnett’s test showed that all groups presented a significant increase in MAT oxidation (*p* < 0.001). At 12 weeks of life, there was no difference between groups, and MAT oxidation was similar to that seen in deficient one-week-old animals (Figure 6A). A significant negative linear correlation was observed between lung GSH and oxidized MAT in liver for 1-week-old animals (Lung GSH = (−3.38 × MAT_Oxidized/Total_) + 15.21; *r*^2^ = 0.38; *p* < 0.001). This relationship was not significant in the 12-week-old animals (Lung GSH = (0.96 × MAT_Oxidized/Total_) + 45.76; *r*^2^ = 0.002; *p* = 0.80) (Figure 6B).

## 4. Discussion

This study supports the hypothesis by demonstrating that a diet deficient in antioxidants during the neonatal period can induce a long-term modification of redox metabolism in lungs. Diets deficient in vitamin C or cysteine, independently of each other, caused a drop in glutathione levels during the duration of these diets and an increase 11 weeks after cessation of these diets (12 weeks of life). However, only cysteine deficiency modulated gene transcription of *Keap1*, *Gsr* and *Glrx*—lower than the control at 1 week of age and higher at 12 weeks—while also modulating the long-term protein levels of GR and GRX.

This study also demonstrates the reliance of lung glutathione levels and redox metabolism on the oxidation status of liver MAT, an enzyme needed for glutathione synthesis (Figure 7). This hepatic metabolism is highly affected by vitamin C and cysteine deficiencies. This is one of the few studies to demonstrate the interorgan effect of protein oxidation over redox metabolism and the only one, to our knowledge, to demonstrate it following neonatal deficiencies in vitamin C or in cysteine. This relationship has been demonstrated for liver MAT activity and plasma GSH levels [17] but not liver MAT and lung GSH.

Interestingly, vitamin C and cysteine deficiencies have different effects. Therefore, we cannot attribute these consequences solely to their effect on decreased GSH levels. At one week of life, vitamin C deficiency leads to a decrease in lung GSH and ascorbate, an oxidation of redox potential of glutathione and the oxidation of liver MAT. The decrease in lung ascorbate levels illustrates the physiological deficiency caused by the diet. The same decrease was already observed before in liver and in other similar animal models [7,24,28]. Given that ascorbate can reduce S-nitrosylated cysteines [27,29] and sulfenic acid protein adducts [30], and that MAT can be inactivated by these posttranslational modifications [18,31,32], it is likely that any oxidation of cysteine residues in MAT will remain oxidized as ascorbate levels are not enough to reduce them. Oxidized MAT leads to lower availability of cysteine for glutathione synthesis in liver. Since the lungs are dependent on liver glutathione synthesis [33,34,35], the lower hepatic synthesis leads to lower lung GSH (Figure 7).

On the other hand, neonatal cysteine deficiency leads to a decrease in total glutathione (GSH and GSSG) and ascorbate levels. Cysteine is the limiting substrate in GSH synthesis [19,20]; therefore, it was expected to observe a decrease in GSH and GSSG levels. However, no changes in redox potential were observed in the CD group. Simultaneously, we also observed a decrease in *Keap1*, *Gsr* and *Glrx* mRNA levels. Unfortunately, our experimental design could not demonstrate a significant relationship between Nrf2 and *Gclm* or *Gclc* expression, two Nrf2 targets [36]. *Gsr* codes for glutathione-disulfide reductase (GR), and *Glrx* codes for glutaredoxin (GRX). These enzymes work together to deglutathionylate cysteinyl residues in proteins. First, glutaredoxin deglutathionylates proteins while forming an internal disulfide in its active site. Afterwards, two molecules of GSH reduce this disulfide, generating GSSG, which in turn is reduced back to GSH by glutathione reductase [37]. Therefore, it is logical that the expression of these genes is regulated similarly. Indeed, the expression of their genes is likely to be regulated by the same mechanism in our model, given the strong positive correlation between them. However, no significant changes in the protein levels of these proteins were observed at 1 week of age. This result represents a dissociation between gene expression and protein translation, and it could be due to oxidative signals that affect epigenetic mechanisms. Indeed, it has been demonstrated that oxidative stress increases DNA methylation [21,22]. It is possible that the lack of glutathione decreases the expression of these genes through epigenetic mechanisms, without affecting protein levels. Would exposure for more than 4 days have induced a change in protein levels?

At 12 weeks of age, no changes in MAT oxidation were observed in animals in vitamin C deficiency groups. However, lung GSH and GSSG levels were 24% and 74% higher, respectively. This seems to be an adaptive response to increased total glutathione following lower glutathione at earlier life. In CD and DD groups at 12 weeks of age, an increase in GSH and GSSG was observed, much like that caused by early life vitamin C deficiency. However, cysteine deficiency seems to lead to additional effects, as it raises *Keap1*, *Gsr* and *Glrx* mRNA levels. These effects are the opposite from those observed at 1 week, which appears to be a possible compensatory long-term effect to regulate redox metabolism. Indeed, this increase in mRNA levels could be tentative to reach homeostasis by increasing the protein levels of these proteins that have been decreased. Given that GSH is a co-substrate of GRX in protein disulfide reduction, an increase in GSH concentration could also be a compensatory mechanism to support GRX activity. The decreased levels of GR and GRX could increase glutathionylation levels in adult lungs. Protein glutathionylation in lungs is associated with airway inflammation and fibrosis, while glutaredoxin seems to be beneficial in ablating inflammatory responses triggered by asthma initiators [38,39,40]. It is therefore possible that these long-term effects of cysteine deficiency—and to a lesser extent, vitamin C deficiency—in the neonatal period could increase the risk of pulmonary conditions such asthma and chronic obstructive pulmonary disease. Indeed, preterm neonates that go through neonatal oxidative stress often develop these conditions at adulthood [1,2,3,4,5,6,7,8,9,10,11]. Future studies should focus on the effect of these neonatal deficiencies on the development of chronic lung diseases in adulthood.

The study highlights several limitations in our knowledge of the impact of antioxidant deficiency occurring during the neonatal period. It was expected to observe a decrease in lung ascorbate levels in animals fed a diet deficient in vitamin C. However, a similar observation in cysteine deficiency animals was surprising and remains unexplained. As for the impact of vitamin C deficiency on the level of GSH and not on GSSG, leading to an oxidation of the redox potential of glutathione, it was unexpected. The metabolic pathway based on MAT oxidation in the liver could be part of the explanation. Knowing that the prevalence of ascorbate deficiency is high in developed countries [41,42], it becomes very relevant to deepen our knowledge of the metabolic relationship between vitamin C and glutathione. Another limit is the magnitude of the deficiencies. The lack of additive effect of vitamin C and cysteine deficiencies on the levels of GSH and ascorbate observed at one week of life was statistically documented by a significant interaction. The interpretation could be that each deficiency reached a maximum effect, which results in a plateau in doubly deficient animals. This plateau could make it difficult to discriminate the difference in metabolic effects between cysteine and ascorbate deficiency. We also observed that the decreases in the expression of *Gsr* and *Glrx* at 1 week of age in the double deficient group are not as severe as the decrease in single deficient animals. This effect has been observed in other models of long-term effects of early life insults. One study tested the combination of the effects of malnutrition during the gestation period with the effects of leptin treatment on the offspring. Leptin, when administered during the neonatal period, reduces consumption and mimics malnutrition. Although gestational malnutrition and leptin have very similar effects in modulating the expression of genes related to energy metabolism, mitochondrial function and immune response, the combination of the two inhibits this programming [43]. However, the underlying mechanisms are not yet understood. Although the data suggest a lower deglutathionylation activity, this needs to be demonstrated. Unfortunately, the present study does not allow us to do so. Given the importance of this finding, further study should be undertaken to confirm our findings by measuring glutathionylation activity in relation to lung disease.

This is the first study, to our knowledge, to describe the evolution of protein oxidation between neonatal life and early-adulthood. In this study, we assessed MAT, which is more reduced early in life in controls. This suggests that neonatal proteins are more susceptible to protein oxidation, which makes early life oxidative stress a very important programming factor. We have already demonstrated the effects of early life oxidative stress over the development of neonatal diseases [1,4,7,21,26,44,45], while several adult diseases are also associated with protein oxidation [38,39,46,47,48,49]. At 12 weeks of age, many of the variables that were decreased early in life were increased in similar proportions at 12 weeks, as a rebound effect. A similar effect is observed in animals and humans who went through protein and energy deficiencies early in life [50,51,52]. This effect was named “The Thrifty Phenotype Hypothesis” by Baker [53]. This is the first time an effect similar to that described in the thrifty phenotype hypothesis is observed for redox metabolism outcomes, notably GSH, GSSG and the mRNA levels of *Keap1*, *Gsr* and *Glrx*. Studies that induced neonatal vitamin C or glutathione deficiencies have been done, but they have not focused on biochemical parameters [54,55]. Energy metabolism is directly coupled to redox metabolism. GSH/GSSG and NADPH are similar to ATP/ADP and NADH, and in both cases, glucose is the main source of energy to generate NAD[P]H. It is possible that the same mechanisms that drive the changes observed in the thrifty phenotype hypothesis also drive the changes observed in our neonatal antioxidant deficiency model. These include but are not limited to perturbations in epigenetics [21,56], organ structure [9,57] and accelerated aging [7,11]. Given how oxidative stress can modulate each of these factors [7,9,21], they are good candidate mechanisms that would explain this effect in our model.

## 5. Conclusions

We demonstrate in this study the importance of adequate nutrition at neonatal life. Vitamin C and cysteine deficiencies lead to liver MAT oxidation and lung oxidative stress. Cysteine deficiency also modulates the expression of genes involved in deglutathionylation at early life and adulthood and programs less deglutathionylation at 12 weeks. This mechanism could be the link between neonatal oxidative stress and adult lung diseases.

## Figures and Tables

**Figure 1 antioxidants-12-01361-f001:**
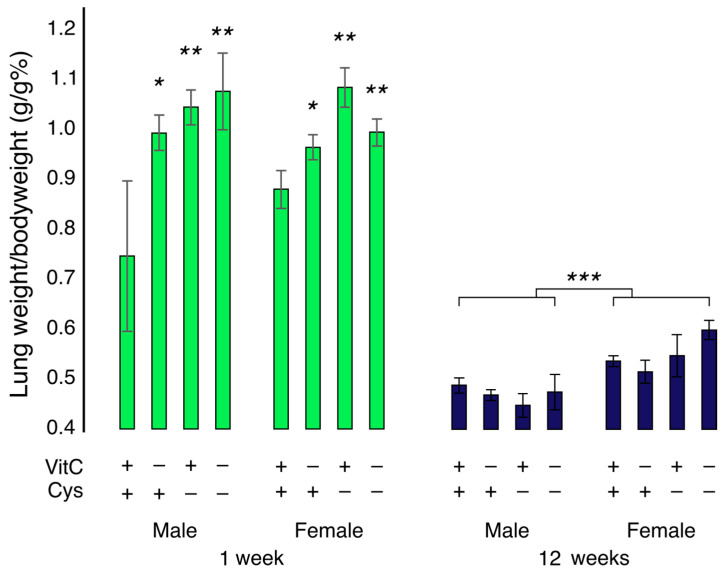
Lung weight per bodyweight (g/g %) in male and female guinea pigs at 1-week and 12-week. The percentage of lung weight was higher at 1 week of life in deficient animals from both sexes. At 12 weeks of life, no difference in the percentage of lung weight was observed among groups, but females had a higher percentage of lung weight per bodyweight. An interaction between sex and cysteine deficiency was observed, as this deficiency decreases the percentage of lung weight/bodyweight in males and increases it in females at 12 weeks of life (F_Cys × sex(1,24)_ = 5.0, *p* < 0.05). Mean ± S.E.M.; *: *p* < 0.05; **: *p* < 0.01; ***: *p* < 0.001. n = 3–5/group.

**Figure 2 antioxidants-12-01361-f002:**
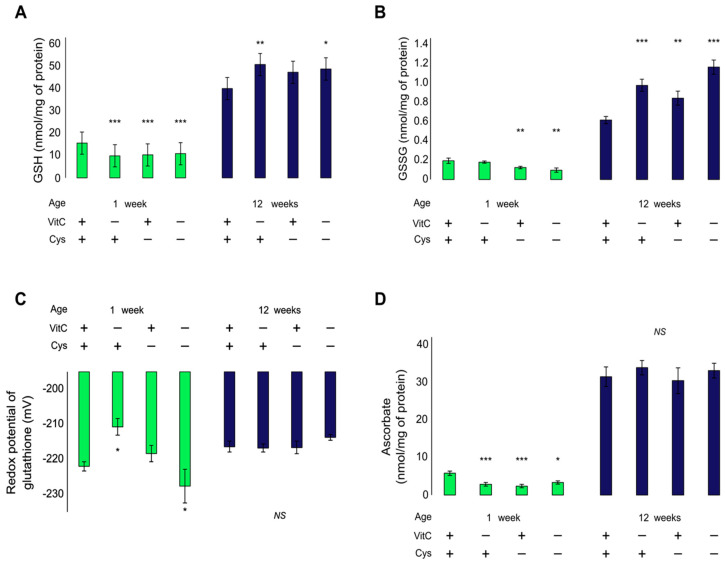
Lung GSH, GSSG, redox potential of glutathione and ascorbate in 1-week and 12-week animals. (**A**): GSH levels were decreased in vitamin C and cysteine-deficient animals at 1 week of life. At 12 weeks of life, early-life vitamin C deficiency increased lung GSH levels. (**B**): GSSG was decreased by cysteine deficiency at 1 week of life and increased by both vitamin C and cysteine deficiencies at 12 weeks. The long-term effects of vitamin C and cysteine deficiencies are independent. (**C**): Redox potential of glutathione was oxidized (increased) by vitamin C deficiency at 1 week and reduced (decreased) by the combination of both deficiencies. No effect of single cysteine deficiency was observed. No effect was observed at 12 weeks. (**D**): Lung ascorbate levels were decreased in all deficient groups at 1 week. No effect was observed at 12 weeks. Mean ± S.E.M.; NS: statistically non-significant; *: *p* < 0.05; **: *p* < 0.01; ***: *p* < 0.001. n = 7–8/group.

**Figure 3 antioxidants-12-01361-f003:**
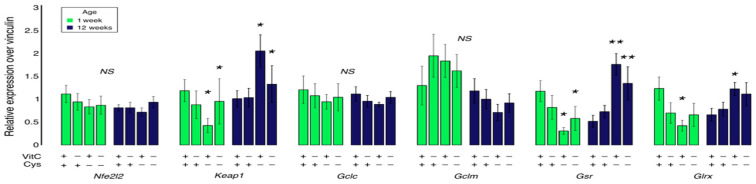
Relative expression (mRNA levels) in lungs of *Nfe2l2*, *Keap1*, *Gclc*, *Gclm*, *Gsr* and *Glrx* in 1-week and 12-week animals. No statistical effect was observed for *Nfe2l2*, *Gclc* or *Gclm*. At 1 week, cysteine deficiency decreased *Keap1*, *Gsr* and *Glrx* gene expression. At 12 weeks, these same genes were significantly more expressed in animals who had a cysteine deficiency early in life. Mean ± S.E.M.; NS: statistically non-significant. *: *p* < 0.05; **: *p* < 0.01. n = 7–8/group.

**Figure 4 antioxidants-12-01361-f004:**
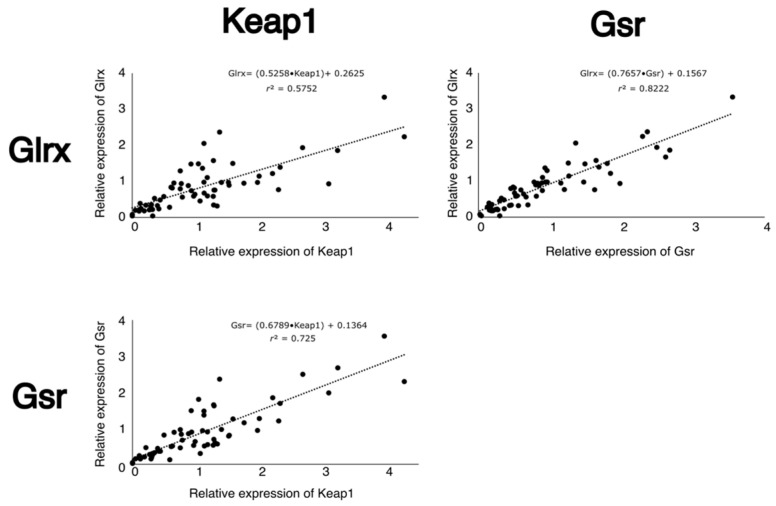
Correlation between the three genes altered by cysteine deficiency (*Keap1*, *Gsr* and *Glrx*). All three genes correlate positively with each other, suggesting a single mechanism of modulation. Keap1 × Glrx (Glrx_mRNA_ = 0.5258 × (Keap1_mRNA_) + 0.2625; *r*^2^ = 0.58; *p* < 0.001); Keap1 × Gsr (Gsr_mRNA_ = 0.6789 × (Keap1_mRNA_) + 0.1364; *r*^2^ = 0.73; *p* < 0.001); Gsr × Glrx (Glrx_mRNA_ = 0.7657 × (Gsr_mRNA_) + 0.1567; *r*^2^ = 0.82; *p* < 0.001). n = 63.

**Figure 5 antioxidants-12-01361-f005:**
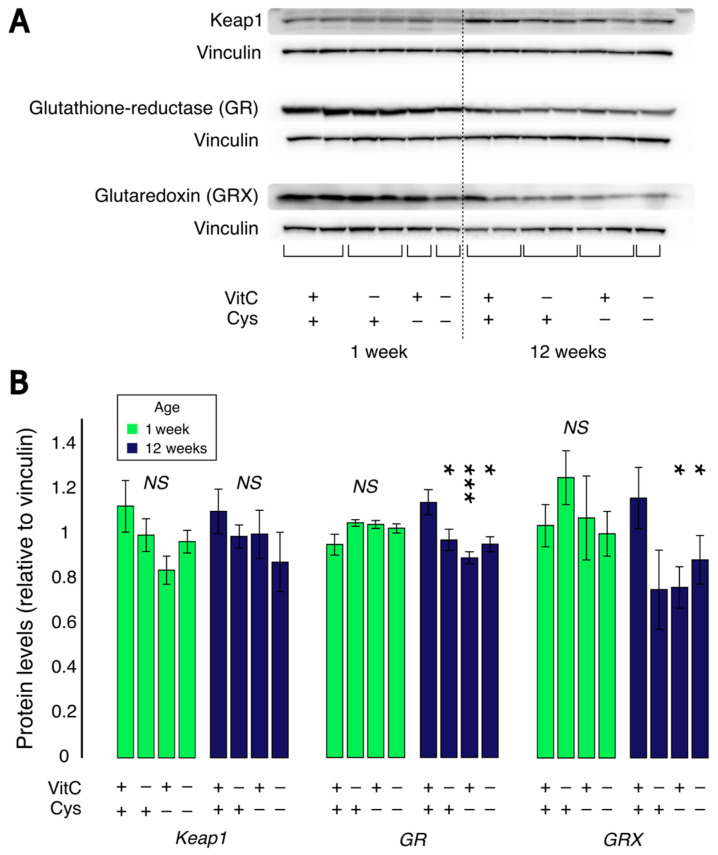
Protein levels of Keap1, glutathione-reductase (GR) and glutaredoxin (GRX) in lungs. (**A**): Representative Western blot images from Keap1, GR, GRX and vinculin as a reference protein. (**B**): The relative protein levels of Keap1, GR and GRX. No significant changes were observed in protein levels of all three proteins at 1 week of age. At 12 weeks, a decrease in GR protein levels was observed in all previously deficient groups (*p* < 0.05). GRX levels were decreased at 12 weeks of age in animals having received a cysteine deficient diet at neonatal age (*p* < 0.05). Mean ± S.E.M.; NS: statistically non-significant; *: *p* < 0.05; ***: *p* < 0.001. n = 7–8/group.

**Figure 6 antioxidants-12-01361-f006:**
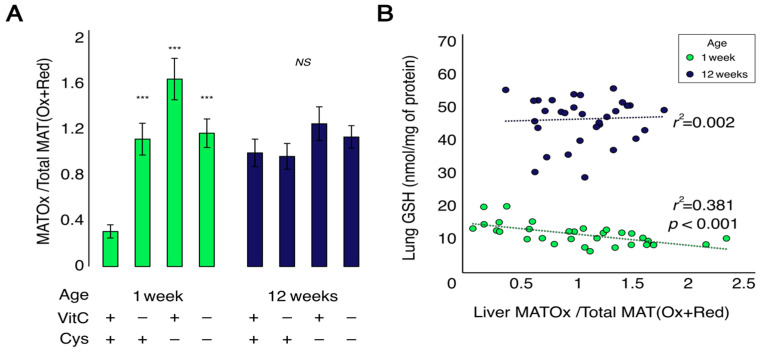
Liver methionine-adenosyltransferase (MAT) oxidation and its correlation with lung GSH levels. (**A**): The oxidation of MAT cysteine residues (MATOx) was increased in both vitamin C and cysteine-deficient animals at 1 week of age. No significant effect was observed at 12 weeks. (**B**): Liver MAT oxidation negatively correlates with lung GSH levels in 1-week-old animals but not in 12-week-old animals. Mean ± S.E.M.; NS: statistically non-significant; ***: *p* < 0.001. n = 7–8/group.

**Figure 7 antioxidants-12-01361-f007:**
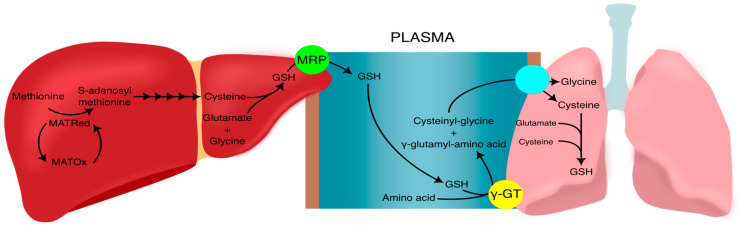
The pathway of cysteine, from methionine in the liver to GSH in the lung. In the liver, the state of oxidation of MAT determines its activity. When in a reduced state (MATRed), MAT catalyzes the limiting reaction in the transmethylation pathway, which ultimately leads to the synthesis of cysteine from methionine. Cysteine is the limiting substrate in GSH synthesis. GSH is exported from the liver by MRP in the hepatocyte membrane. In plasma, GSH is transported to extra-hepatic tissues containing γGT, such as the lungs. γGT breaks down GSH into cysteinylglycine and γ-glutamyl-amino acid, which are further broken down by membrane dipeptidases and imported by amino acid transporters, both represented by the blue circle. Once inside the pneumocytes, the amino acids can act as substrates for GSH synthesis. The oxidation of liver MAT blocks this interorgan pathway. GSH: glutathione; γGT: γ-glutamyl transpeptidase; MAT: methionine-adenosyltransferase; MRP: multidrug resistance protein.

**Table 1 antioxidants-12-01361-t001:** Primer sequences and its concentrations used in RT-qPCR experiments.

Gene and Ensemble Code	Coded Protein	Primer	Primer Sequence	Concentration Used (nM)
*Nfe2l2* *(ENSCPOG00000031261)*	Nuclear factor erythroid 2–related factor 2 (Nrf2)	Forward	GCTAGATGAAGAGACAGGGGA	500
		Reverse	ACAAATGGGAATGTTTCTGCCA	
*Keap1* *(ENSCPOG00000038105)*	Kelch-like ECH-associated protein 1 (Keap1)	Forward	TGCTACAACCCCATGACCAA	400
		Reverse	ACCAAGTGCCACTCGTCC	
*Gclc* *(ENSCPOG00000008314)*	Glutamate—cysteine ligase catalytic subunit (GCLC)	Forward	TGGGGAGAAGTACAACGACA	400
		Reverse	GGCATCATCCAGGTCGATCT	
*Gclm* *(ENSCPOG00000037468)*	Glutamate—cysteine ligase modifier subunit (GCLM)	Forward	CCTAGACAAAACACAGTTGGAGC	300
		Reverse	AGCTTCTTGGAAACTTGCTTCA	
*Gsr* *(ENSCPOG00000005290)*	Glutathione-disulfide reductase (GR)	Forward	ATGTTGACTGCCTGCTCTGG	300
		Reverse	TGCGTAGATGCCTTTGACACT	
*Glrx* *(ENSCPOG00000040776)*	Glutaredoxin (GRX)	Forward	GATCCTCAGTCAGTTGCCCT	500
		Reverse	CGATGCAGTCTCTCCCGATG	
*Vcl* *(ENSCPOG00000005058)*	Vinculin (VCL)	Forward	ACCACAACTCCCATCAAGCT	500
		Reverse	ACCACAACTCCCATCAAGCT	

## Data Availability

All data are included in the article.

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
