# Peer review of "Disturbances of the Lung Glutathione System in Adult Guinea Pigs Following Neonatal Vitamin C or Cysteine Deficiency"

_antioxidants, 2023, doi:10.3390/antiox12071361_

Round 1

Reviewer 1 Report (Previous Reviewer 1)

The authors have improved the article which is now suitable for publication. Please correct glutathionylation « statiuu » in the abstract.

Author Response

Manuscript Antioxidants-2465222, title:  "Disturbances of the lung glutathione system in adult guinea pigs following neonatal vitamin C or cysteine deficiency"
Authors: Vitor Teixeira, Ibrahim Mohamed  and Jean-Claude Lavoie *

Answers to the Reviewer 1, third round. 
Comment (C), Answer (A)

C1: Please correct glutathionylation « statiuu » in the abstract.
A: Thank you, the word has been corrected.

Reviewer 2 Report (Previous Reviewer 2)

In this new version, the authors have added new experiments which better illuminate the subject they are investigating.

Although corresponding measurements of GSH transferase activity and the oxidation state of Keep1 should shed more light on the responsible mechanisms, the overall quality of the work has been satisfactory upgraded.

Author Response

Manuscript Antioxidants-2465222, title:  "Disturbances of the lung glutathione system in adult guinea pigs following neonatal vitamin C or cysteine deficiency"
Authors: Vitor Teixeira, Ibrahim Mohamed  and Jean-Claude Lavoie *

Answers to the Reviewer 2, third round. 
Comment (C), Answer (A)

C1: Although corresponding measurements of GSH transferase activity and the oxidation state of Keep1 should shed more light on the responsible mechanisms, the overall quality of the work has been satisfactory upgraded. 
A: We agree with the reviewer. It would have been interesting to expand the data on the multiple aspects of glutathione metabolism. This was not the aim of the present work. As always, a research project not only brings new knowledge, but also very frequently generates new questions, which in turn lead to new avenues of research.  

Reviewer 3 Report (New Reviewer)

1. What is the main question addressed by the research?

The authors tried to answer the question whether the neonatal short-time dietary deficiency in vitamin C or cysteine or both might affect the lung antioxidant-glutathione system in adult animals. As a research model the guinea pig study was chosen.

2. Do you consider the topic original or relevant in the field? Does it

address a specific gap in the field?

Although the topic is not entirely original as the literature provides tons of knowledge about antioxidant deficiency in several organs, there is still a gap about specific mechanisms related to dietary antioxidant withdrawal in vulnerable young organisms and what are consequences of that status in elder ages. The experimental schema is to be admired.

3. What does it add to the subject area compared with other published

material?

I suppose that the literature is quite rich in knowledge about the topic but the authors gave us the opportunity to see more details because the experimental schema is brilliant and it provides opportunity to answer “big” questions.

4. What specific improvements should the authors consider regarding the

methodology? What further controls should be considered?

The construction of the study is very right.

5. Are the conclusions consistent with the evidence and arguments presented

and do they address the main question posed?

The conclusions are right and justified by the main text. But, I need more information about the differed consequences between the single deficiency and the double (DD) deficiency. Expand conclusions in the abstract and at the end of the text.

6. Are the references appropriate?

The references were used in an appropriate manner.

7. Please include any additional comments on the tables and figures.

Abstract and the text: “mRNA levels of kelch-like ECH-associated protein 1 (Keap1), glutathione-reductase (Gsr) and glutaredoxin-1 (Glrx) were significantly lower only in CD.” – please try to explain why such observations were not noted in the DD group?

Introduction: at the end of that paragraph a clear hypothesis must be added.

Why did you include sex as an contributing factor to the outcomes? If you mean that the sex is so important please add some info to the Introduction section.

Author Response

Manuscript Antioxidants-2465222, title:  "Disturbances of the lung glutathione system in adult guinea pigs following neonatal vitamin C or cysteine deficiency"
Authors: Vitor Teixeira, Ibrahim Mohamed  and Jean-Claude Lavoie *

Answers to the Reviewer 3, third round. 
Comment (C), Answer (A)

C1: Abstract and the text: “mRNA levels of kelch-like ECH-associated protein 1 (Keap1), glutathione-reductase (Gsr) and glutaredoxin-1 (Glrx) were significantly lower only in CD.” – please try to explain why such observations were not noted in the DD group?
A: The words “but not in DD”, have been added in the 11th line of the abstract, and we have added at the end of page 12 (Discussion section) the following: 
    We also observed that the decreases in the expression of Gsr and Glrx at 1 week of age in the double deficient group is not as severe as the decrease in single deficient animals. This effect has been observed in other models of long-term effects of early life insults. One particular study tested the combination of the effects of malnutrition during the gestation period with the effects of leptin treatment on the offspring. Leptin, when administered during the neonatal period, reduces consumption and mimics malnutrition. Although gestational malnutrition and leptin have very similar effects in modulating the expression of genes related to energy metabolism, mitochondrial function, and immune response, the combination of the two inhibits this programming [43]. However, the underlying mechanisms are not yet understood.

C2: Introduction: at the end of that paragraph a clear hypothesis must be added.
A: The following hypothesis has been added: We hypothesized that an oxidative stress in neonatal life leads to an increase in antioxidant defenses at adulthood in guinea pigs.

C3: Why did you include sex as an contributing factor to the outcomes? If you mean that the sex is so important please add some info to the Introduction section.
A: The granting agency advocates that health research include an assessment of the potential impact of gender on outcomes. Although there is no evidence that glutathione metabolism is related to gender in guinea pigs, the study did take this into account. This is important because glutathione metabolism in human neonates differs according to sex ( Lavoie JC, Tremblay A. Sex-specificity of oxidative stress in newborn leading to a personalized antioxidant nutritive strategy. Antioxidants (Basel). 2018 Mar 27;7(4):49. doi: 10.3390/antiox7040049). The lack of difference between males and females in guinea pigs suggests that this aspect of glutathione metabolism differs between humans and guinea pigs. Although not an essential aim of the study, these data should be published. Thus, the following sentence has been added at the end of the introduction: : Although males and females were used in this study, sex was not found to be a contributing factor to the assessed outcomes.

This manuscript is a resubmission of an earlier submission. The following is a list of the peer review reports and author responses from that submission.

Round 1

Reviewer 1 Report

Oxidative stress may be responsible for bronchopulmonary dysplasia in premature infants fed with parenteral nutrition. Teixeira et al. have thus studied in guinea pigs the influence of an oral diet complete or deficient in vitamin C (VCD), Cystein (CD) or both (DD) for 4 days. The authors have evaluated at 1 week and 12 weeks of age lung weight, glutathione (oxidized and reduced), ascorbate, the expression of Keap1, Nfe2l2, Gclc, Gclm, Gsr and Glrx mRNA as well as liver methionine-adenosyltransferase (oxidized/total) at the protein level. The main results obtained by the authors show that glutathione and mRNA from glutathione-related enzymes are differently affected during CD and VCD, supporting the hypothesis that a deficiency in antioxidants such as Cystein and Vitamin C during the neonatal period may induce a long-term redox perturbation in adults. The work is of interest, the article is well written and no additional experiments are needed in my opinion but several important issues should be addressed as stated below.

In the title of the article and throughout the manuscript, the authors focused on the studied enzymes as being glutathionylation enzymes.  For instance, the authors highlight Gsr and Glrx as enzymes which activities lead to protein deglutathionylation (from line 300). The glutathione system is not solely involved in protein (de)glutathionylation. I would not highlight so strongly the glutathionylation enzymes but also the glutathione system and its contributions related to oxidative stress for instance. This is especially important in my opinion since you did not study if gluathionylation of lung proteins were affected in your work.

Lines 23 and 254: In the abstract and discussion, the authors mention a “permanent” change (in glutathione metabolism). The evaluations were performed at 1 week and 12 weeks, therefore we cannot really talk about a “permanent” change. “Long-term” change would be more appropriate.

Lines 26 and 32: DOHaD and COPD. Although these are common abbreviations, since they are used only once, you may want to give the full terms rather than the abbreviations. Check also carefully for the abbreviations the first time they are used (e.g. TCEP, DMF…).

Paragraph 2.2.2 RT-qPCR (from Line 99): you did not mention how you used the Vinculin (household gene I suppose) for the RT-qPCR. You should give information related to the normalization and other possible relevant information.

Lines 141-142: you need to provide more information related to the Western blot protocol (primary antibody dilution, secondary antibody…).

Paragraph 3.1; 3.2; 3.3 and corresponding Figures. You should mention in the titles (paragraphs and figures) that this is in lung.

Title of paragraph 3.4: you should mention that this is in liver in the title of the paragraph.

Figure 4 : “Keap1” and “Gsr” should be moved to below the x axis.

Line 285: change “reduce of S-nitrosylated” to “reduce S-nitrosylated”.

The authors should remind the readers towards the end of the discussion or conclusion that the perturbation of the expression of the glutathione-related enzymes and transcription factors were observed at the mRNA level and that further studies will be required to see whether the corresponding activities are indeed affected.  This comment stands also for the hypothetical perturbation of protein glutathionylation which will also have to be investigated in the future.

Lines 323-325: you link higher Keap1 mRNA to a possible enhanced cytoplasmic retention of Nrf2. However, you have not checked if Keap1 is indeed induced at the protein level, and how this would possibly affect the nucleocytoplasmic transfer of Nrf2. In my opinion I would either be more careful about this statement or, even better, remove the sentence.

Line 380-381 (Supplementary Materials): this suggests that you have some to provide (Figure S1, Table S1…) which is not the case. You may want to simply indicate that there is none.

Author Response

Comment (C), Answer (A)

Reviewer 1:
Oxidative stress may be responsible for bronchopulmonary dysplasia in premature infants fed with parenteral nutrition. Teixeira et al. have thus studied in guinea pigs the influence of an oral diet complete or deficient in vitamin C (VCD), Cystein (CD) or both (DD) for 4 days. The authors have evaluated at 1 week and 12 weeks of age lung weight, glutathione (oxidized and reduced), ascorbate, the expression of Keap1, Nfe2l2, Gclc, Gclm, Gsr and Glrx mRNA as well as liver methionine-adenosyltransferase (oxidized/total) at the protein level. The main results obtained by the authors show that glutathione and mRNA from glutathione- related enzymes are differently affected during CD and VCD, supporting the hypothesis that a deficiency in antioxidants such as Cystein and Vitamin C during the neonatal period may induce a long-term redox perturbation in adults. 
The work is of interest, the article is well written and no additional experiments are needed in my opinion but several important issues should be addressed as stated below. 

C 1:   In the title of the article and throughout the manuscript, the authors focused on the studied enzymes as being glutathionylation enzymes. For instance, the authors highlight Gsr and Glrx as enzymes which activities lead to protein deglutathionylation (from line 300). The glutathione system is not solely involved in protein (de)glutathionylation. I would not highlight so strongly the glutathionylation enzymes but also the glutathione system and its contributions related to oxidative stress for instance. This is especially important in my opinion since you did not study if gluathionylation of lung proteins were affected in your work. 

A:     We agree with you. The Title has been modified by removing the reference to glutathionylation. Therefore the new title is now:  Increased lung glutathione system in adult guinea pigs following neonatal vitamin C or cysteine deficiency.

C 2:   Lines 23 and 254: In the abstract and discussion, the authors mention a “permanent” change (in glutathione metabolism). The evaluations were performed at 1 week and 12 weeks, therefore we cannot really talk about a “permanent” change. “Long-term” change would be more appropriate. 
A:     We agree. In the abstract and Discussion sections, the term “permanent” has been replaced with “Long-term”. In the Discussion section, the first sentence has been adjusted by replacing “The study confirms the hypothesis that a …”  with “The study supports the hypothesis by demonstrating that a …”.

C 3:   Lines 26 and 32: DOHaD and COPD. Although these are common abbreviations, since they are used only once, you may want to give the full terms rather than the abbreviations. Check also carefully for the abbreviations the first time they are used (e.g. TCEP, DMF...).  
A:     This is done in the new version of the manuscript.

C 4:   Paragraph 2.2.2 RT-qPCR (from Line 99): you did not mention how you used the Vinculin (household gene I suppose) for the RT- qPCR. You should give information related to the normalization and other possible relevant information.   

A:     The description of the method used to calculate relative expression with vinculin as a housekeeping gene has been added in section 2.2.1, at the end of the second paragraph as follows: Vinculin was used as a housekeeping gene and relative expression was calculated with the ΔΔCt method, where:  ΔCt = (CtGene – Ct Vcl); ΔΔCt = ΔCtExpreimental – ΔCtControl group; and Relative expression = 2– ΔΔCt.

C 5:   Lines 141-142: you need to provide more information related to the Western blot protocol (primary antibody dilution, secondary antibody...).    
A:    The precisions to the method has been added to the manuscript, at the end of the 2.2.3 section as follows :  … (1:1000) for 1h/room temperature, following with goat anti-mouse IgG-HRP antibody HAF007 (R&D Systems, Minneapolis, MN, USA) (1:2500) for 1h/room temperature. Antibodies were diluted in 2.5% w/v skim milk in phosphate-buffer-saline-Tween-20.

C 6:   Paragraph 3.1; 3.2; 3.3 and corresponding Figures. You should mention in the titles (paragraphs and figures) that this is in lung.     
A:    This is done in the new version of the manuscript.

C 7:   Title of paragraph 3.4: you should mention that this is in liver in the title of the paragraph.      
A:    This is done in the new version of the manuscript.

C 8:   Figure 4 : “Keap1” and “Gsr” should be moved to below the x axis. 
A:    This is done in the new version of the manuscript.

C 9:   Line 285: change “reduce of S-nitrosylated” to “reduce S-nitrosylated” 
A:    This is done in the new version of the manuscript.

C 10:   The authors should remind the readers towards the end of the discussion or conclusion that the perturbation of the expression of the glutathione-related enzymes and transcription factors were observed at the mRNA level and that further studies will be required to see whether the corresponding activities are indeed affected. This comment stands also for the hypothetical perturbation of protein glutathionylation which will also have to be investigated in the future. 

A:    You are right. Parts of the Discussion have been edited accordingly.
    In the fourth paragraph, the fifth sentence (the one beginning with “We believe …”) has been deleted. In the following sentence (the one beginning with « Unfortunately”), the word “ increase ” has been replaced with “ relationship between  Nrf2 and ”. In the twelfth sentence (the one beginning with “ Thus, it seems …”) the word “ seems ” has been replaced with “ could be ”. After the fourteenth sentence (the one beginning with “ These defects may…”), two new sentences have been added: “ However, the study was not designed to investigate these enzymes at protein and activity levels. Given the importance of these metabolisms, an additional study should be designed specifically to demonstrate this possibility.”

C 11:   Lines 323-325: you link higher Keap1 mRNA to a possible enhanced cytoplasmic retention of Nrf2. However, you have not checked if Keap1 is indeed induced at the protein level, and how this would possibly affect the nucleocytoplasmic transfer of Nrf2. In my opinion I would either be more careful about this statement or, even better, remove the sentence.  
A:    In the fifth paragraph,  the seventh sentence (the one beginning with “ Higher Keap1 mRNA …”) as well as the eighth sentence, have been deleted. The beginning of the following sentence (“On the other hand,”) have also be deleted while the following has been added at the end of the sentence: “ …, which could be demonstrated, or investigated, by a subsequent study.”

C 12:   Line 380-381 (Supplementary Materials): this suggests that you have some to provide (Figure S1, Table S1...) which is not the case. You may want to simply indicate that there is none. 
A:     Thank you

Reviewer 2 Report

Deficiencies in parenteral nutrition factors administered in premature infants may induce oxidative stress and contribute to the development of adult lung diseases. For this reason, it is important to investigate the relationship of nutrition both with the induction of oxidative stress and with the long-term effects.

In this work of Teixeira et al. examine the short and delayed effects on the lungs of Guinea pigs, following neonatal vitamin C and cysteine deficiency. It has to be noted that this model has been previously extensively used by the same group. In this work however, the authors have focused on redox metabolism and particularly on GSH/GSSG levels and related enzymes, as well as protein glutathionylation in lungs.

Main point

The observations of gene expression changes reported at mRNA level, should also be substantiated by corresponding measurements at the protein level (Western blotting) and mainly with estimation of enzyme activities (if possible). Moreover, additional experiments, which will illustrate direct effects of dietary deficiencies on lung protein glutathionylation (using specific antibodies) are absolutely necessary to support the theory proposed in the manuscript.

Minor point

Page 11 (line 364): The expression “Redox metabolism shares similarities with energy metabolism …” is not accurate, since energy metabolism is directly coupled to redox metabolism. 

Author Response

Comment (C), Answer (A)

Reviewer 2:
Deficiencies in parenteral nutrition factors administered in premature infants may induce oxidative stress and contribute to the development of adult lung diseases. For this reason, it is important to investigate the relationship of nutrition both with the induction of oxidative stress and with the long-term effects. 
In this work of Teixeira et al. examine the short and delayed effects on the lungs of Guinea pigs, following neonatal vitamin C and cysteine deficiency. It has to be noted that this model has been previously extensively used by the same group. In this work however, the authors have focused on redox metabolism and particularly on GSH/GSSG levels and related enzymes, as well as protein glutathionylation in lungs. 

C 1:   The observations of gene expression changes reported at mRNA level, should also be substantiated by corresponding measurements at the protein level (Western blotting) and mainly with estimation of enzyme activities (if possible). Moreover, additional experiments, which will illustrate direct effects of dietary deficiencies on lung protein glutathionylation (using specific antibodies) are absolutely necessary to support the theory proposed in the manuscript. 

A:    We agree with this reviewer. There could be a world between the mRNA and protein activity. A similar comment was also given by the first reviewer. We believe that the response produced for the first reviewer (in italics) can respond to this comment. We have edited the part of the discussion section that was too assertive on the activity of these proteins. We believe that the new version of the manuscript better reflects the results obtained. We also emphasize the importance of studying the translation of mRNA observations to the activity of these proteins. 
    In the fourth paragraph, the fifth sentence (the one beginning with “We believe …”) has been deleted. In the following sentence (the one beginning with « Unfortunately”), the word “ increase ” has been replaced with “ relationship between  Nrf2 and ”. In the twelfth sentence (the one beginning with “ Thus, it seems …”) the word “ seems ” has been replaced with “ could be ”. After the fourteenth sentence (the one beginning with “ These defects may…”), two new sentences have been added: “ However, the study was not designed to investigate these enzymes at protein and activity levels. Given the importance of these metabolisms, an additional study should be designed specifically to demonstrate this possibility.”

C2: Page 11 (line 364): The expression “Redox metabolism shares similarities with energy metabolism ...” is not accurate, since energy metabolism is directly coupled to redox metabolism. 
A:     Thank you. In the new version, the sentence “Energy metabolism is directly coupled to redox metabolism.” replaces the sentence “Redox metabolism shares similarities with energy metabolism.”

Round 2

Reviewer 2 Report

As can be presumed from authors response, the authors are not willing to make any additional experimental efforts to strengthen their article. However, without fulfilling, at list partly, the main requirements addressed in the initial report, I don't think the article can be published at its present form. 

Author Response

Manuscript Antioxidants-2111940, new title:  "Disturbances of the lung glutathione system in adult guinea pigs following neonatal vitamin C or cysteine deficiency"
Previous title: "Increased pulmonary glutathione and expression of glutathionylation enzymes in adult guinea pigs following neonatal vitamin C or cysteine deficiency"
Authors: Vitor Teixeira, Ibrahim Mohamed  and Jean-Claude Lavoie *

Answers to the Reviewer 2, second round
Comment (C), Answer (A)

Reviewer 2:
Deficiencies in parenteral nutrition factors administered in premature infants may induce oxidative stress and contribute to the development of adult lung diseases. For this reason, it is important to investigate the relationship of nutrition both with the induction of oxidative stress and with the long-term effects. 

In this work of Teixeira et al. examine the short and delayed effects on the lungs of Guinea pigs, following neonatal vitamin C and cysteine deficiency. It has to be noted that this model has been previously extensively used by the same group. In this work however, the authors have focused on redox metabolism and particularly on GSH/GSSG levels and related enzymes, as well as protein glutathionylation in lungs. 

C:   The observations of gene expression changes reported at mRNA level, should also be substantiated by corresponding measurements at the protein level (Western blotting) and mainly with estimation of enzyme activities (if possible). Moreover, additional experiments, which will illustrate direct effects of dietary deficiencies on lung protein glutathionylation (using specific antibodies) are absolutely necessary to support the theory proposed in the manuscript. 

A:      Thank you for this suggestion. Western bolts have been performed for Keap1, GR and GRX.  The results were surprising as they went in a different direction that their mRNA levels. This has led to a better understanding of the potential impact of these neonatal nutritive deficiencies on the glutathione system later in life. These results could be important for understanding several lung deceases observed in adults. Of course, the link between our findings and disease remains to be confirmed by further investigations, including measurements of glutathionylation activity in the lung. Unfortunately, we were unable to measure these activities. 

    The text has therefore been modified on several levels. 

1) The title has been adapted to the new reality of the manuscript. The title "Increased pulmonary glutathione and expression of glutathionylation enzymes in adult guinea pigs following neonatal vitamin C or cysteine deficiency", has been replaced by "Disturbances of the lung glutathione system in adult guinea pigs following neonatal vitamin C or cysteine deficiency".

2) The last two sentences of the Abstract are now  "Keap1, Gsr and Glrx mRNA were increased, while glutathione-reductase and glutaredoxin proteins were lower in CD, favoring a higher glutathionylation statiuu. Both neonatal deficiencies result in a long-term change in glutathione metabolism that could contribute to lung diseases development."

3) At the end of the last sentence of the Introduction, we have added "… which is accompanied by a long-term decrease in deglutathionylation enzymes, such as glutaredoxin and glutathione-reductase."

4) In Materials and Methods section, a new sub-section has been added : "2.2.3. Keap1, GR and GRX protein levels."

5) In Results section, a new sub-section has been added : "3.4. Protein levels of Keap1, GR and GRX », including a new Figure (#5) with part A for Western blot illustrations and part B for results expressed as an histogram. 

6) In Discussion section, the changes were mainly by 

i) adding "while also modulating the long-term protein levels of GR and GRX. "  at the end of the first paragraph; 

ii) change the end of the fourth paragraph which is now "However, no significant changes in the protein levels of these proteins were observed at 1 week of age. This result represents a dissociation between gene expression and protein translation, and it could be due to oxidative signals that affect epigenetic mechanisms. Indeed, it has been demonstrated that oxidative stress increases DNA methylation [21,22]. It is possible that the lack of glutathione decreases the expression of these genes through epigenetic mechanisms, but the changes in protein levels. Would exposure for more than 4 days have induced a change in protein levels? ";  

iii) change the end of the fifth paragraph which is now: "Indeed, this increase in mRNA levels could be tentative to reach homeostasis by increasing the protein levels of these proteins that has been decreased. Given that GSH is a co-substrate of GRX in protein disulfide reduction, an increase in GSH concentration could also be a compensatory mechanism to support GRX activity. The decreased levels of GR and GRX could increase glutathionylation levels in adult lungs. Protein glutathionylation in lungs is associated with airway inflammation and fibrosis, while glutaredoxin seems to be beneficial in ablating inflammatory responses triggered by asthma initiators [38–40]. It is therefore possible that these long-term effects of cysteine deficiency – and to a lesser extent, vitamin C deficiency – in the neonatal period could increase the risk of pulmonary conditions such asthma and chronic obstructive pulmonary disease. Indeed, preterm neonates that go through a neonatal oxidative stress often develop these conditions at adulthood[1–11]. Future studies should focus on the effect of these neonatal deficiencies on the development of chronic lung diseases in adulthood." ; 

iv) adding in the sixth paragraph this new limitation of our study: "Although the data suggest a lower deglutathionylation activity, this needs to be demonstrated. Unfortunately, the present study does not allow us to do so. Given the importance of this finding, further study should be undertaken to confirm our findings by measuring glutathionylation activity in relation to lung disease."  ;

v) The conclusion has been adjusted : "We demonstrate in this study the importance of adequate nutrition at neonatal life. Vitamin C and cysteine deficiencies lead to liver MAT oxidation and lung oxidative stress. Cysteine deficiency also modulates the expression of genes involved in deglutathionylation at early life and adulthood, and programs less deglutathionylation at 12 weeks. This mechanism could be the link between neonatal oxidative stress and adult lung diseases."

Thanks to the reviewer for helping us improve the manuscript.  
